# Does the Manipulation Process Matter? RITA: Reasoning Composite Image Manipulations via Reversely-Ordered Incremental-Transition Autoregression

## Abstract

Image manipulations often entail a complex manipulation process, comprising a series of editing operations to create a deceptive image, exhibiting **sequentiality** and **hierarchical** characteristics. However, existing IML methods remain manipulation-process-agnostic, directly producing localization masks in a one-shot prediction paradigm without modeling the underlying editing steps. This one-shot paradigm compresses the high-dimensional compositional space into a single binary mask, inducing severe **dimensional collapse**, thereby creating a fundamental mismatch with the intrinsic nature of the IML task.

To address this, we are the first to reformulate image manipulation localization as a conditional sequence prediction task, proposing the **RITA** framework. RITA predicts manipulated regions layer-by-layer in an ordered manner, using each step's prediction as the condition for the next, thereby explicitly modeling temporal dependencies and hierarchical structures among editing operations.

To enable training and evaluation, we synthesize multi-step manipulation data and construct a new benchmark **HSIM**. We further propose the **HSS** metric to assess sequential order and hierarchical alignment. Extensive experiments show RITA achieves SOTA on traditional benchmarks and provides a solid foundation for the novel hierarchical localization task, validating its potential as a general and effective paradigm. The code and dataset will be publicly available.

## 1 Introduction

Most existing Image Manipulation Localization (IML) methods Chen et al. (2021); Wang et al. (2022); Kwon et al. (2022); Liu et al. (2022a); Guillaro et al. (2023); Zhu et al. (2025) adopt a one-shot prediction paradigm. In this paradigm, a model inputs an image and directly outputs a unified binary mask (Fig. 1(a)) to identify all potentially manipulated regions, answering the question of *where* the manipulation occurred. This paradigm indeed offers enhanced interpretability over simple binary classification and has achieved notable success on current benchmarks. Nevertheless, it fundamentally oversimplifies the intrinsic nature of complex, localized image manipulations in real-world scenarios.

In real-world scenarios, image manipulations are rarely accomplished in a single step but are instead constructed through a sequence of ordered, localized editing operations. For example, an attacker might first copy-paste a background patch to cover an object, then implant new content directly onto this pasted area, and finally perform touch-ups. Such a process exhibits two distinct characteristics: 1) **Sequentiality**, as edits must be applied in a specific temporal order, and 2) **Hierarchy**, as later operations often build upon prior edits, forming spatial and semantic dependencies. Thus, the prevailing one-shot paradigm in IML only outputs a single mask that answers *Where* manipulation occurred, collapsing the high-dimensional compositional space of sequential and hierarchical edits into a single binary mask. This **dimensional collapse** forces the model to discard crucial temporal and structural cues, leaving it to learn only coarse spatial distributions. As a result, this prediction paradigm is fundamentally mismatched with the IML task's intrinsic nature.

Figure 1: Comparison between (a) the standard one-shot localization and (b) our RITA framework.

To resolve this problem, we are the first to reformulate the task of content manipulation localization from a one-shot prediction problem to a **conditional sequence prediction** problem. Accordingly, we design a novel autoregressive localization framework, termed **RITA (Reversely-ordered Incremental-Transition Autoregression)**. As illustrated in Fig. 1(b), the core idea of RITA is to progressively and explicitly answer *how* a manipulation is constructed by predicting manipulation regions layer by layer in an ordered sequence, conditioning each prediction on the input image and the output from the immediately preceding step. This design explicitly models the temporal dependencies and hierarchical structures among editing steps, thereby naturally and effectively disentangling the layered composition of complex content manipulations and resolving the dimensional mismatch with the task's intrinsic nature.

To enable effective training and evaluation for this new paradigm, we design a tree-structured reverse sampling strategy to construct a synthetic dataset that simulates the multi-step hierarchical editing process of real-world manipulations. Additionally, we annotate the **Hierarchical Sequential Image Manipulation (HSIM)** dataset, the first real-world benchmark with multi-step manipulated paths. Furthermore, we introduce the **Hierarchical Sequential Score (HSS)**, a metric that measures the accuracy of predicted sequences in terms of both sequential order and hierarchical alignment.

At the same time, our paradigm remains compatible with traditional one-shot datasets by treating them as a special case of a two-step process. Extensive experiments on traditional datasets confirm strong generalization and competitive efficiency, validating that our sequence modeling approach is a broadly applicable and effective localization paradigm, not limited to hierarchical structures.

Our main contributions are summarized as follows:

- We are the first to reformulate the manipulation localization paradigm from one-shot prediction to sequence prediction and propose **RITA**, the first framework to explicitly model the temporal order and hierarchical structure of manipulated operations.

- We construct **HSIM**, the first real-world dataset that reflects multi-step manipulation processes. In addition, we design a tree-based sampling strategy to generate a synthetic dataset that models hierarchical dependencies.

- We introduce **HSS**, a new evaluation metric designed for hierarchical manipulation localization, which evaluates both structural and sequential predictions.

- Extensive experiments demonstrate that RITA is not only effective on the new hierarchical task but also exhibits superior performance in generalization, robustness, and efficiency on traditional benchmarks.

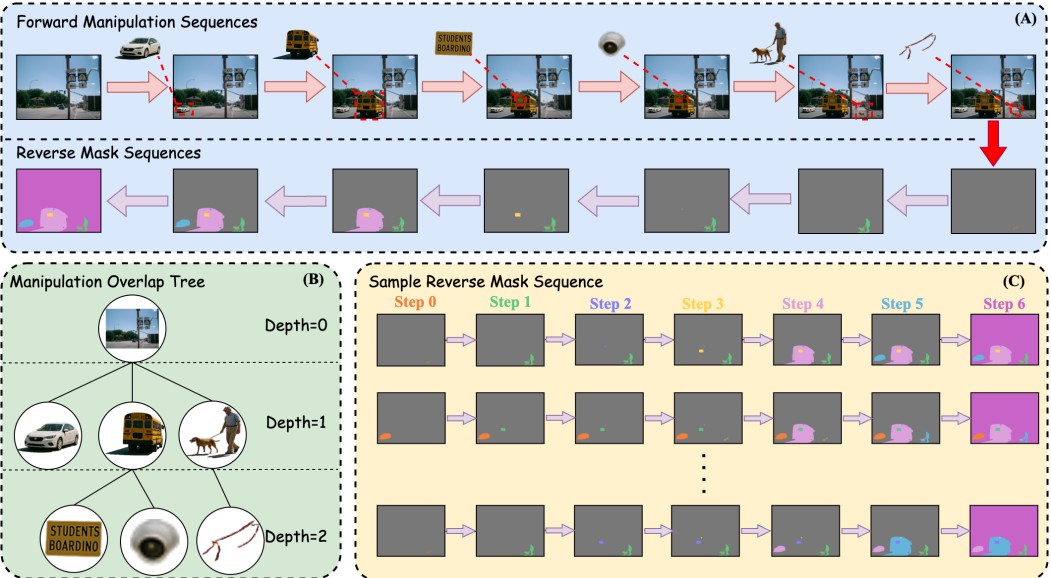

Figure 2: Illustration of the synthetic multi-step manipulated process. (A) Sequential application of localized manipulations to an image, which depicts both the macroscopic manipulated process and the prediction of a corresponding mask at each step. (B) Organization of the manipulated regions into a hierarchical tree structure, where the figure shows the constructed tree for the current example. (C) Reverse-order sampling from the manipulation tree to construct mask sequences, where manipulated areas are accumulated step by step. Gray regions denote padding used to align the mask sequences.

## 2 RELATED WORKS

Most IML methods Wu et al. (2019); Chen et al. (2021); Kwon et al. (2022); Guillaro et al. (2023); Wang et al. (2022); Zhu et al. (2025); Liu et al. (2022a); Su et al. (2025); Ma et al. (2023) follow a unified paradigm: given an image, the model outputs a single binary mask covering all manipulated regions. This design collapses the inherently multi-dimensional nature of manipulation into a single-step prediction, ignoring the sequential and structural aspects of the manipulation process.

SAFIRE Kwon et al. (2025) emphasizes source separation but similarly overlooks temporal dependencies within edits from the same source. This further reduces multi-step manipulations into a flattened representation, showing that source-level cues alone are insufficient to capture the full dimensionality of manipulation required for detailed forensic analysis.

## 3 METHOD

**Problem Formulation: Conditional Sequence Prediction.** We reformulate manipulation localization as a conditional sequence prediction task, progressively predicting a sequence of masks $(M_1, \ldots, M_T)$ to reveal the manipulation history. While the space of all mask sequences is the vast Cartesian product $\mathcal{M}^T$ (where $\mathcal{M} = \{0,1\}^{H \times W}$), valid sequences must obey the **progressive containment property**:

$$M_t \subseteq M_{t+1} \quad \text{for all } t \in \{1, \ldots, T-1\}. \tag{1}$$

This constraint dictates that manipulated regions only accumulate, meaning our goal is to predict a monotonically increasing path of masks. As shown in Figure 2, our approach is built upon three key steps: (A) sequential localized manipulations to form composite manipulations, (B) hierarchical organization of manipulated regions into a tree structure, and (C) reverse-order sampling of manipulation masks from the tree to form autoregressive training sequences.

We first describe dataset construction and evaluation metrics (Sec. 3.1), formalize the conditional Markov process underlying conditional sequence prediction (Sec. 3.2), and present our model design (Sec. 3.3).

## 3.1 DATASET CONSTRUCTION AND EVALUATION METRICS

To enable fair evaluation on both traditional benchmarks and real-world sequence scenarios, we unify all datasets under a sequence prediction paradigm. Specifically, for traditional one-shot datasets such as CASIAv2, we reformulate them into two autoregressive steps, while we also construct new datasets with multiple manipulated operations to reflect realistic manipulation paths. We also introduce a new metric to evaluate this new scenario. This unified design allows consistent training and evaluation across both traditional and hierarchical manipulation cases.

### 3.1.1 SEQUENCE MANIPULATED DATASET CONSTRUCTION

To train and evaluate our model on hierarchical manipulations, we synthesize a sequence manipulation dataset by extending benchmarks like CASIAv2. We apply a series of random copy-move operations, where each new region corresponds to a **node** in a tree. The spatial relationships between these operations, such as nesting and adjacency, naturally form a hierarchical structure. This hierarchy is formally captured as a **manipulation tree**, which serves as the real-world representation for the manipulation process.

**Tree-Structured Manipulation Representation.** As illustrated in Fig. 2(B), we represent the spatial hierarchy of manipulated regions as a tree $\mathcal{T} = (V, E)$, where each node $v \in V$ corresponds to a distinct manipulated region $R(v)$. The tree is constructed dynamically based on a clear hierarchical rule: the parent of a new region is the deepest node in the existing tree that contains this new region.

$$\text{parent}(v_j) = \underset{v \in V' \text{ s.t. } R(v_j) \subseteq R(v)}{\text{argmax}} (\text{depth}(v), \text{Area}(R(v))).$$ (2)

Here, $V'$ denotes the set of pre-existing nodes. An edge $(v_i \rightarrow v_j)$ is then added to $E$, defining the containment hierarchy. The root node $v_{\text{root}}$ represents the full image domain, and the tree correctly simulates the partial order of operations.

**Sequential Path Sampling.** A single manipulation tree $\mathcal{T}$ defines the hierarchical structure of edits, but not a unique linear order. This ambiguity means that multiple valid sequential paths can represent the same set of hierarchical edits. To capture this diversity for training, we generate $N$ distinct sequences by simulating plausible "undo" trajectories (Fig. 2(C)). Each path, denoted $\mathcal{P}_i = (v_1, \ldots, v_{T_i})$, is constructed via **reverse-sampling**—a process of iteratively selecting and removing a random leaf node from the tree. At each step $k$, a node is drawn uniformly from the current set of leaf nodes:

$$v_k \sim \text{Uniform}(\text{Leaves}(\mathcal{T} \setminus \{v_1, \ldots, v_{k-1}\})).$$ (3)

From this path, we construct the final accumulated mask sequence $\{M_t^{(i)}\}$. The mask at each step $t$ is the union of the regions from the first $t$ nodes in the path:

$$M_t^{(i)} = \text{Union}_{j=1}^t R(v_j).$$ (4)

This node-centric construction guarantees the progressive containment property ($M_t^{(i)} \subseteq M_{t+1}^{(i)}$), which is essential for autoregressive modeling. It implies that at any step $t$, the model operates on the revealed content within $M_t^{(i)}$, while the unrevealed areas are masked out using padding.

**Hierarchical Sequential Image Manipulation Dataset.** To evaluate under realistic conditions, we construct a real-world composite dataset. Starting from a pool of authentic photographs, we generate stepwise manipulation plans with GPT-4o and execute the edits using GPT-Image-1. AI editing often introduces *undesired artifacts*, where regions outside the intended manipulated area are inadvertently modified or distorted Costanzino et al. (2025). To mitigate this, we perform targeted manual refinements so that only the intended regions are altered while the surrounding content remains intact. After sampling, HSIM contains **2,442 valid manipulation paths**, each supplied with stepwise masks for autoregressive evaluation.

### 3.1.2 Compatibility with Conventional Datasets

For existing datasets such as CASIAv2, which only provide a single-step binary mask, we design a simple yet effective decomposition strategy to make them compatible with our autoregressive paradigm. Specifically, the image is split into two regions: the manipulated region $M_{\text{tamper}}$ and the authentic region $M_{\text{auth}} = 1 - M_{\text{tamper}}$. This enables us to reinterpret the binary annotation as a **two-step** incremental sequence.

The mask sequence is constructed as:

$$M_t = \begin{cases} M_{\text{tamper}} \cup M_{padding}, & t = 1, \\ M_{\text{full}}, & t = 2, \end{cases} \tag{5}$$

where $M_{padding}$ denotes a placeholder region for alignment.

At $t = 1$, the model focuses solely on the manipulated region, while the authentic region is masked out. At $t = 2$, the model predicts the full image domain, including both manipulated and authentic regions. This design ensures that one-shot datasets can be seamlessly incorporated into our sequence autoregressive paradigm, maintaining consistency across all training and evaluation settings.

### 3.1.3 Hierarchical Sequential Score

To evaluate predictions on our tree-structured data, we must account for the non-unique manipulation orders. We therefore evaluate a single prediction against all $N$ valid paths that can be derived from a manipulation tree $\mathcal{T}$ (Fig. 2(C)), combining structural alignment and length consistency.

**Structure Matching.** To handle discrepancies in length between predicted sequences and ground-truth paths, we utilize MonotonicMatch, a dynamic programming algorithm (detailed in the Appendix A.1). For each ground-truth path, this algorithm computes an optimal monotonic, non-decreasing alignment with the predicted sequence, maximizing the stepwise F1 score between them.

Let the predicted mask sequence be $P \in \mathbb{R}^{T_p \times H \times W}$ with $T_p$ steps, and the ground-truth path set be $M \in \mathbb{R}^{N \times T_g \times H \times W}$ containing $N$ sampled paths each with $T_g$ steps. For each path $M_i$, we compute

$$\text{F1}_{\text{match}} = \max_{i=1,\ldots,N} \text{MonotonicMatch}(P, M_i), \tag{6}$$

**Final Score with Length Consistency.** To penalize mismatched step counts between prediction length $T_p$ and ground-truth length $T_g$, we apply a length penalty:

$$\lambda(T_p, T_g) = \exp\left(-\alpha \frac{(T_p - T_g)^2}{\max(T_g, 1)}\right), \tag{7}$$

where $\alpha = 0.1$ controls the penalty strength. The final score is defined as

$$\text{Score} = \lambda(T_p, T_g) \times \text{F1}_{\text{match}}. \tag{8}$$

This metric, termed the **Hierarchical Sequential Score (HSS)**, jointly evaluates sequential accuracy and structural alignment, ensuring fair assessment across multiple valid manipulation paths.

### 3.2 Conditional Markov Formulation

We formulate mask prediction as a conditional sequence prediction task. In general, predicting the next mask $M_{t+1}$ would depend on the input image $I$ and the all history of previous masks $(M_0, \ldots, M_t)$. However, due to the **progressive containment property** ($M_t \subseteq M_{t+1}$), the current mask $M_t$ acts as a sufficient statistic, encapsulating all spatial information from the preceding steps. This insight allows us to simplify the dependency to a **first-order Markov assumption**:

$$P(M_{t+1} \mid I, M_t, \ldots, M_0) = P(M_{t+1} \mid I, M_t). \tag{9}$$

Consequently, the full sequence probability decomposes autoregressively, reducing our task to learning the one-step transition model $P(M_{t+1} \mid I, M_t)$:

$$P(M_1, \ldots, M_T \mid I) = \prod_{t=0}^{T-1} P(M_{t+1} \mid I, M_t). \tag{10}$$

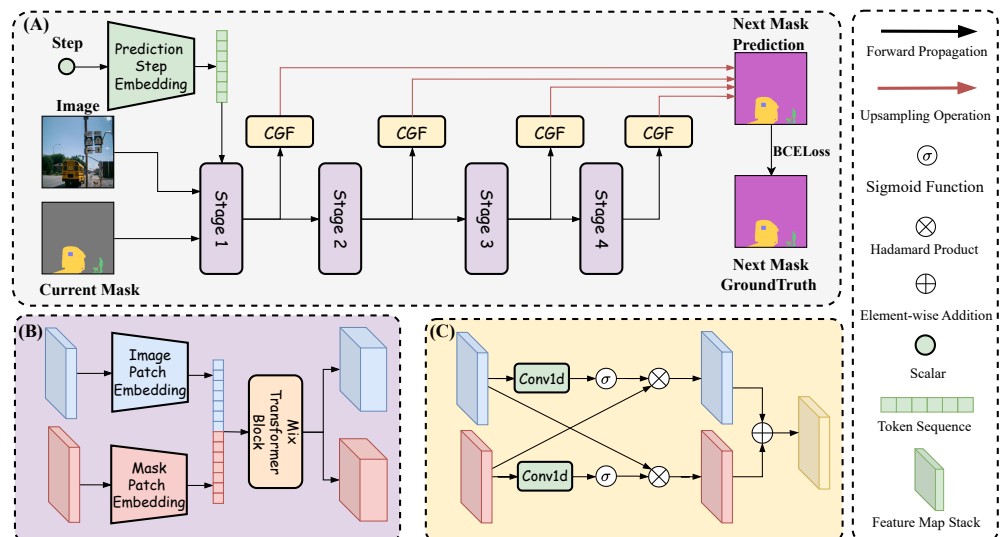

Figure 3: Overall architecture of our proposed framework. Given the manipulated input image, the current mask, and the current step embedding, the model progressively encodes and fuses multi-scale image and mask features through attention interaction and TransitionGatedFusion modules, enabling step-by-step decomposition of composite manipulations.

## 3.3 MODEL DESIGN

Our framework follows a conditional first-order Markov structure 3.2, where the prediction of the next mask $M_{t+1}$ depends only on the current mask $M_t$ and the input image $I$.

**Architecture Overview.** As illustrated in Figure 3(A), the model takes as input the manipulated image $I$, the current mask $M_t$, and the step index $s_t$. A learnable embedding maps $s_t$ to a vector $E_t$, which is injected into the first-stage mask features to provide temporal conditioning. The inputs are processed by a multi-scale encoder, cross-modal fusion modules, and a decoder that outputs the next mask prediction $M_{t+1}$.

**Multi-Scale Encoder.** The encoder design is shown in Figure 3(B). The image and mask are separately processed into hierarchical feature maps:

$$\{F_I^{(l)}\}_{l=1}^4 = \text{Stage}_I(I), \quad \{F_M^{(l)}\}_{l=1}^4 = \text{Stage}_M(M_t), \tag{11}$$

where $l$ denotes the stage index. At the first stage, the mask features are augmented with the broadcasted step embedding:

$$F_M^{(1)} \leftarrow F_M^{(1)} + E_t. \tag{12}$$

At each stage, image and mask tokens are concatenated, passed through a Mix Transformer block, and split again into two streams, enabling joint reasoning while preserving modality-specific information.

**Transition Gated Fusion.** To capture manipulation cues, we introduce a TransitionGatedFusion module at each scale (Figure 3(C)). Given image and mask features, we compute:

$$G_I^{(l)} = \sigma(W_M F_M^{(l)}), \tag{13}$$

$$G_M^{(l)} = \sigma(W_I F_I^{(l)}), \tag{14}$$

$$\hat{F}^{(l)} = F_I^{(l)} \odot G_M^{(l)} + F_M^{(l)} \odot G_I^{(l)}, \tag{15}$$

where $W_I$ and $W_M$ are 1×1 convolutions, $\sigma(\cdot)$ is the sigmoid function, and $\odot$ denotes element-wise multiplication. This cross-gating mechanism ensures that image features are modulated by mask context and vice versa, focusing attention on evolving manipulated regions.

**Decoder and Mask Prediction.** Fused features from all scales are upsampled and concatenated with mask features:

$$Z_t = \text{Concat}(\hat{F}^{(1)}, \hat{F}^{(2)\uparrow}, \hat{F}^{(3)\uparrow\uparrow}, \hat{F}^{(4)\uparrow\uparrow\uparrow},$$
$$F_M^{(1)}, F_M^{(2)\uparrow}, F_M^{(3)\uparrow\uparrow}, F_M^{(4)\uparrow\uparrow\uparrow}) \uparrow\uparrow, \tag{16}$$

where $\uparrow$ indicates progressive upsampling by a factor of 2 at each stage. A $1\times1$ convolutional classifier then outputs per-pixel class probabilities:

$$p_\theta^{(t)} = \text{Softmax}(W_{\text{cls}} * Z_t). \tag{17}$$

**Training Objective.** We train the model autoregressively with **teacher forcing**. Initialization uses $M_0$, an all-padding mask denoted `START`. At each step $t$, the model predicts $M_{t+1}$ conditioned on $(I, M_t^{\text{GT}})$ and is supervised by per-pixel cross-entropy against $M_{t+1}^{\text{GT}}$. The ground-truth sequence ends with an `EOS` mask, which the model learns to emit as the stop signal. The loss is defined as:

$$\mathcal{L}_{\text{CE}}^{(t)} = -\frac{1}{HW} \sum_{x,y} \log p_\theta^{(t)}[M_{t+1}^{\text{GT}}(x,y)], \tag{18}$$

where $p_\theta^{(t)}[\cdot]$ is the predicted probability for the ground-truth class at pixel $(x, y)$.

**Edge Supervision Loss.** To refine boundary localization, we add an auxiliary edge loss. The incremental manipulated region between two consecutive ground-truth masks is

$$\Delta M_{t+1} = M_{t+1}^{\text{GT}} - M_t^{\text{GT}}. \tag{19}$$

Its boundary $\text{Boundary}(\Delta M_{t+1})$ is extracted using morphological dilation. The predicted mask $M_{t+1}^{\text{Pred}}$ is supervised only along these boundary pixels:

$$\mathcal{L}_{\text{Edge}}^{(t)} = \text{BCE}\big(\text{Boundary}(M_{t+1}^{\text{Pred}}),\ \text{Boundary}(\Delta M_{t+1})\big). \tag{20}$$

The total training loss at step $t$ is

$$\mathcal{L}^{(t)} = \mathcal{L}_{\text{NMP}}^{(t)} + \beta\, \mathcal{L}_{\text{Edge}}^{(t)}, \tag{21}$$

with $\beta$ balancing next-mask prediction and edge refinement.

**Inference.** At inference, the process begins with $M_0$ initialized as a special `START` mask. At each step $t$, the model predicts a distribution

$$p_\theta^{(t)}(x,y) = f_\theta(I, M_t)(x,y), \tag{22}$$

and derives the next mask as

$$M_{t+1}(x,y) = \arg\max_k\ p_\theta^{(t)}(x,y,k), \tag{23}$$

where $k \in \{0, \ldots, K-1\}$ includes all semantic classes and a special `EOS` class. The autoregressive process terminates once the proportion of pixels predicted as `EOS` exceeds a predefined threshold $\tau$ (set to 95% in our experiments), or when the maximum step count $T_{\text{max}}$ is reached.

# 4 EXPERIMENTS

In this section, we report experiments that (1) demonstrate compatibility with the traditional one-shot IML paradigm and (2) evaluate performance under newly constructed sequence manipulation scenarios. We further include focused ablations and backbone-controlled comparisons to substantiate generalization and the contribution of each component.

## 4.1 EXPERIMENTS SETUP

All experiments were conducted on eight NVIDIA RTX 3090 GPUs using PyTorch 2.6 with CUDA version 12.4. Input images were uniformly resized to a resolution of 512×512. During training, a batch size of 20 was used, while a batch size of 12 was employed for evaluation. The learning rate followed a cosine decay schedule Loshchilov & Hutter (2017), starting at 1e-4 and decreasing to a minimum of 5e-7. We adopted the AdamW optimizer Loshchilov & Hutter (2019) with a weight decay of 0.05 to alleviate overfitting.

## 4.2 Evaluation Metric

For the existing traditional manipulation scenarios, we follow the training and evaluation protocol of the fully aligned IMDLbenco Ma et al. (2024), using the binary F1 score as the performance metric. Meanwhile, for the sequence manipulation scenarios, we adopt the previously introduced metric 3.1.3 to assess the model's performance under complex manipulation conditions.

## 4.3 Traditional manipulation Scenarios

### 4.3.1 Performance Comparison

In this subsection, following Subsection 3.1.2, we adapt existing one-shot manipulation datasets into a **two-step incremental sequence** for our RITA framework. In contrast, existing baselines such as MVSS-Net Chen et al. (2021), PSCC-Net Liu et al. (2022a), Cat-Net Kwon et al. (2022), TruFor Guillaro et al. (2023), and Mesorch Zhu et al. (2025) are evaluated in their original one-shot prediction paradigm. All methods are compared on both source-aligned (CASIA v1 Dong et al. (2013)) and cross-source datasets (Coverage Wen et al. (2016), Columbia Hsu & Chang (2006), NIST16 Guan et al. (2019), IMD2020 Novozamsky et al. (2020), CocoGlide Guillaro et al. (2023), Autosplice Jia et al. (2023)), following the MVSS and the Cat-Net training protocol Ma et al. (2024).

Table 1: Comparison of F1 scores across datasets. The table is divided into two parts: the upper part reports results under the **MVSS protocol**, while the lower part follows the **CAT-Net protocol**. The best result per column is marked in red, and the second-best is underlined.

| Model | Source-Aligned | Cross-Source | | | | | | | Overall Avg |
|---|---|---|---|---|---|---|---|---|---|
| | CASIAv1 | Coverage | Columbia | NIST16 | IMD2020 | CocoGlide | Autosplice | Cross-Source Avg | |
| MVSS-Net | 0.534 | 0.259 | 0.386 | 0.246 | 0.279 | 0.291 | 0.294 | 0.293 | 0.327 |
| CAT-Net | 0.581 | 0.296 | 0.584 | 0.269 | 0.273 | 0.290 | 0.354 | 0.361 | 0.378 |
| PSCC-Net | 0.378 | 0.231 | 0.604 | 0.214 | 0.235 | 0.227 | 0.652 | 0.377 | 0.363 |
| TruFor | 0.721 | 0.419 | 0.865 | 0.324 | 0.322 | 0.205 | 0.393 | 0.421 | 0.464 |
| Mesorch | 0.740 | 0.326 | 0.726 | 0.343 | 0.269 | 0.162 | 0.249 | 0.346 | 0.402 |
| RITA(Ours) | 0.596 | 0.472 | 0.785 | 0.375 | 0.357 | 0.446 | 0.623 | 0.537 | 0.545 |
| MVSS | 0.583 | 0.482 | 0.740 | 0.336 | – | 0.443 | 0.385 | 0.477 | 0.495 |
| CAT-Net | 0.808 | 0.427 | 0.915 | 0.252 | – | 0.410 | 0.387 | 0.478 | 0.533 |
| PSCC | 0.630 | 0.448 | 0.884 | 0.346 | – | 0.474 | 0.551 | 0.541 | 0.555 |
| Trufor | 0.818 | 0.457 | 0.885 | 0.348 | – | 0.283 | 0.393 | 0.473 | 0.531 |
| Mesorch | 0.840 | 0.586 | 0.890 | 0.392 | – | 0.450 | 0.402 | 0.544 | 0.593 |
| RITA(Ours) | 0.705 | 0.561 | 0.956 | 0.447 | – | 0.493 | 0.644 | 0.620 | 0.634 |

As shown in Table 1, our method achieves strong results across datasets. It obtains the best F1 on most cross-source sets and consistently ranks top or second under both MVSS and CAT-Net protocols. While not overfitting to source-specific artifacts (Source-Aligned F1 not the highest), it delivers the best Cross-Source Avg and Overall Avg. Additional quantitative results are in Appendix A.3.1, and robustness experiments are presented in Appendi A.2.

### 4.3.2 Ablation Study

To evaluate the contribution of each module, we performed ablation experiments by removing or adding components. As shown in Table 2(a), performance drops occur in all cases, confirming the necessity of each design. In particular, removing the Gate Fusion module causes the sharpest decline, highlighting its role in feature integration. We also tested adding a DCT-based feature extractor, but this resulted in a large performance drop, consistent with the result in IMDLBenco Ma et al. (2024).

### 4.3.3 Parameters and FLOPs

We conducted a comparative analysis of all models with a batch size of 1 and an input resolution of 512×512, measuring both parameter count and FLOPs. As shown in Table 2(b), our method achieves the lowest parameter count among recent approaches (except PSCC-Net, which trades extremely few parameters for very high FLOPs) and the best efficiency with only 95.993 GFLOPs, while maintaining best performance.

Table 2: (a) Ablation study of different components (Average F1 score). (b) Comparison of model size and FLOPs under input resolution $512 \times 512$ and batch size 1.

(a) Ablation Study

| Variant | Average | Drop |
|---|---|---|
| **Full Model** | **0.634** | – |
| w/o Edge Supervision | 0.628 | -0.006 |
| w/o Step Embedding | 0.618 | -0.016 |
| w/o Cross Gate Fusion | 0.513 | -0.121 |
| w/ DCT Extractor | 0.402 | -0.202 |

(b) Model Size and FLOPs

| Model | Parameters (M) | FLOPs (G) |
|---|---|---|
| MVSS-Net | 150.528 | 171.008 |
| PSCC-Net | 3.668 | 376.832 |
| CAT-Net | 116.736 | 136.216 |
| TruFor | 68.697 | 236.544 |
| Mesorch | 85.754 | 124.928 |
| Ours | 55.567 | 95.993 |

### 4.4 SEQUENCE MANIPULATION SCENARIOS

Using synthetic and real-world data, we train and evaluate our method under realistic multi-step manipulation scenarios. The synthetic dataset is derived from CASIAv2 through tree-structured copy-move augmentation 3.1.1, where each manipulation tree is decomposed into multiple valid manipulation paths. This process generates 3,680 composite images, with 16,161 autoregressive mask steps produced per sampled path, using a fixed random seed to ensure reproducibility.

For real-world evaluation, we adopt the HSIM dataset introduced in Section 3.1.1. From HSIM, we derive 2,442 valid manipulation paths, which serve as the test set for evaluating autoregressive localization performance using the HSS metric. We conduct two types of experiments. First, we

Table 3: Ablation and baseline comparison on multi-step tampering. (a) Effect of sampled path count. (b) Results of AR variants with different backbones under 3-path training.

(a) Effect of Training Sample Count

| Sample Count | Score | F1$_{match}$ | #Masks |
|---|---|---|---|
| 1 | 0.458 | 0.467 | 16,161 |
| 2 | 0.472 | 0.485 | 32,322 |
| 3 | **0.488** | **0.495** | **48,483** |
| 4 | 0.466 | 0.473 | 64,644 |

(b) AR Reformulations of Existing Backbones

| Model | Score | F1$_{match}$ |
|---|---|---|
| **Segformer** | 0.458 | 0.467 |
| ConvNeXt | 0.449 | 0.460 |
| SwinTransformer | 0.433 | 0.456 |
| Resnet | 0.381 | 0.407 |

perform a controlled ablation 3(a) to assess the impact of varying the number of sampled training paths per manipulation tree. We find that performance improves with more samples, peaking at 3, after which over-sampling may introduce noise and redundancy. Second, we evaluated four different backbones with comparable parameter scales to power our RITA framework: SegFormer-B3 Xie et al. (2021), ConvNeXt-Base Liu et al. (2022b), Swin Transformer-Base Liu et al. (2021), and ResNet-101 He et al. (2016). The results are detailed in Table 3(b). The comparison clearly shows that SegFormer-B3 provides the best performance. Additional quantitative results are presented in Appendix A.3.2.

## 5 CONCLUSION

In this work, we introduce a paradigm shift in image manipulation localization, moving from one-shot prediction to ordered sequence prediction. We identify **dimensional collapse** in existing methods as a key limitation and propose **RITA**, the first autoregressive framework that breaks down manipulations layer-by-layer. Supported by our new **HSIM** dataset and **HSS** metric, our method effectively captures the sequential and structural nature of complex edits.

Our experiments show that the sequential approach works better. RITA performs very well on hierarchical localization tasks and provides a solid baseline for future work. It also outperforms all current state-of-the-art methods on traditional benchmarks. By explaining *how* manipulations happen, our work also offers deeper forensic insight beyond just *where* they are. We hope this inspires future research on understanding and defending against advanced image forgeries.

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

# A   APPENDIX

## A.1   STRUCTURE MATCHING WITH DYNAMIC PROGRAMMING

A core challenge in evaluating the predicted path is that its length ($T_p$) may differ from the ground-truth path's length ($T_g$). A simple frame-by-frame comparison is therefore inadequate. To address this, we introduce **MonotonicMatch**, a dynamic programming algorithm designed to find the optimal alignment between the two sequences.

The goal of MonotonicMatch is to identify a monotonic, non-decreasing mapping between the steps of the predicted sequence and the ground-truth sequence. This mapping is "optimal" in that it maximizes the cumulative stepwise F1 score along the alignment path. First, we compute a pairwise F1 score matrix $F \in \mathbb{R}^{T_p \times T_g}$, where each element $F[i, j]$ represents the F1 score between the $i$-th predicted mask and the $j$-th ground-truth mask. Then, a dynamic programming table is populated to find the path that yields the highest average F1 score, enforcing the monotonic constraint. The detailed procedure is outlined in Algorithm 1.

---

**Algorithm 1:** MonotonicF1Match($P, M$)

---

**Input:** Predicted sequence $P = \{P_1, \ldots, P_{T_p}\}$
Ground-truth sequence $M = \{M_1, \ldots, M_{T_g}\}$
**Output:** Average F1 score of the optimal monotonic alignment
```
// 1.  Compute pairwise similarity matrix
```
Initialize matrix $F \in \mathbb{R}^{T_p \times T_g}$;
**for** $i = 1, \ldots, T_p$ **do**
    **for** $j = 1, \ldots, T_g$ **do**
        $F[i, j] \leftarrow \text{F1\_score}(P_i, M_j)$;

```
// 2.  Find the optimal cumulative score via Dynamic
       Programming
```
Initialize DP table $D \in \mathbb{R}^{T_p \times T_g}$ to store maximum cumulative scores;
**for** $i = 1, \ldots, T_p$ **do**
    **for** $j = 1, \ldots, T_g$ **do**
        **if** $i = 1$ **then**
            $D[i, j] \leftarrow F[i, j]$ ;           `// Base case:  first predicted step`
        **else**
            `// Find max score from any valid previous alignment`
            $max\_prev\_score \leftarrow \max_{1 \le k \le j} D[i-1, k]$;
            $D[i, j] \leftarrow F[i, j] + max\_prev\_score$;

```
// 3.  Extract and normalize the final score
```
$max\_cumulative\_score \leftarrow \max_{1 \le j \le T_g} D[T_p, j]$ ; `// Find the best path's total`
`  score`
**if** $T_p > 0$ **then**
    $F1_{match} \leftarrow max\_cumulative\_score / T_p$;
**else**
    $F1_{match} \leftarrow 0$;
**return** $F1_{match}$;

---

## A.2   ROBUSTNESS

To comprehensively assess the stability of different models under realistic conditions, we consider three representative perturbation families that widely occur in practical imaging scenarios: (i) *Gaussian noise*, which emulates sensor-level corruption and thermal noise during acquisition; (ii) *Gaussian blur*, which mimics defocus and motion blur frequently introduced by imperfect optics or camera shake; and (iii) *JPEG compression*, which reflects storage and transmission artifacts due to lossy coding. For each perturbation family, we progressively increase the perturbation intensity, thereby creating a spectrum of degradation levels that range from mild to severe.

Table 4: **Robustness under common image perturbations.** Entries are *mean Binary F1* on the test set computed under the CAT-Net evaluation protocol. Perturbations include Gaussian noise (standard deviation), Gaussian blur (kernel size), and JPEG compression (quality factor). The rightmost *Average* is the arithmetic mean of the per–condition means within each block (including "None").

| Pertubation | Model | Sandard Deviations | | | | | | | Average |
|---|---|---|---|---|---|---|---|---|---|
| | | None | 3 | 7 | 11 | 15 | 19 | 23 | |
| GaussNoise | MVSS | 0.495 | 0.502 | 0.500 | 0.492 | 0.493 | 0.489 | 0.489 | 0.494 |
| | CAT-Net | 0.533 | 0.512 | 0.500 | 0.484 | 0.473 | 0.462 | 0.454 | 0.488 |
| | PSCC | 0.555 | 0.539 | 0.531 | 0.522 | 0.521 | 0.518 | 0.512 | 0.528 |
| | Trufor | 0.531 | 0.450 | 0.418 | 0.398 | 0.381 | 0.366 | 0.372 | 0.417 |
| | Mesorch | 0.593 | 0.563 | 0.543 | 0.529 | 0.521 | 0.517 | 0.507 | 0.539 |
| | Ours | 0.654 | 0.633 | 0.620 | 0.613 | 0.609 | 0.603 | 0.601 | 0.619 |

| | | Kernel Size | | | | | | | Average |
|---|---|---|---|---|---|---|---|---|---|
| | | None | 3 | 7 | 11 | 15 | 19 | 23 | |
| GaussBlur | MVSS | 0.495 | 0.422 | 0.349 | 0.310 | 0.273 | 0.244 | 0.225 | 0.331 |
| | CAT-Net | 0.533 | 0.487 | 0.458 | 0.429 | 0.417 | 0.402 | 0.392 | 0.445 |
| | PSCC | 0.555 | 0.509 | 0.454 | 0.414 | 0.377 | 0.343 | 0.310 | 0.423 |
| | Trufor | 0.531 | 0.422 | 0.367 | 0.317 | 0.254 | 0.191 | 0.147 | 0.318 |
| | Mesorch | 0.593 | 0.526 | 0.471 | 0.430 | 0.387 | 0.340 | 0.292 | 0.434 |
| | Ours | 0.654 | 0.588 | 0.534 | 0.516 | 0.496 | 0.478 | 0.467 | 0.533 |

| | | Quality Factors | | | | | | | Average |
|---|---|---|---|---|---|---|---|---|---|
| | | None | 100 | 90 | 80 | 70 | 60 | 50 | |
| JpegCompression | MVSS | 0.495 | 0.493 | 0.462 | 0.434 | 0.408 | 0.392 | 0.369 | 0.436 |
| | CAT-Net | 0.533 | 0.547 | 0.522 | 0.495 | 0.484 | 0.480 | 0.474 | 0.505 |
| | PSCC | 0.555 | 0.534 | 0.478 | 0.449 | 0.435 | 0.423 | 0.392 | 0.467 |
| | Trufor | 0.531 | 0.481 | 0.437 | 0.408 | 0.387 | 0.366 | 0.318 | 0.418 |
| | Mesorch | 0.593 | 0.577 | 0.527 | 0.508 | 0.506 | 0.495 | 0.465 | 0.524 |
| | Ours | 0.654 | 0.650 | 0.615 | 0.597 | 0.586 | 0.576 | 0.550 | 0.604 |

Following the CAT-Net evaluation protocol, we compute the mean Binary F1 score for every condition, and summarize the overall performance with a block-wise *Average*, defined as the arithmetic mean of the per–condition means (including the "None" case). This evaluation protocol ensures fairness across methods and allows us to capture the sensitivity of each model to different perturbation strengths.

Results are reported in Table 4. Across all perturbation families, our method consistently achieves the highest averages, obtaining **0.619** under Gaussian noise, **0.533** under Gaussian blur, and **0.604** under JPEG compression. These results represent substantial margins over the strongest baselines (e.g., Mesorch: 0.539/0.434/0.524; CAT-Net: 0.488/0.445/0.505), highlighting the robustness of our approach. More importantly, we observe that prior methods degrade significantly when perturbation severity increases—for example, Trufor exhibits rapid performance drops under blur and noise, and PSCC becomes unstable under low-quality JPEG factors. By contrast, our method maintains comparatively stable scores even at the most extreme settings, such as large noise standard deviations, large blur kernels, and very low JPEG quality.

Taken together, these experiments demonstrate that our model generalizes robustly to realistic degradations and is thus more reliable for deployment in unconstrained environments.

## A.3 QUANTITATIVE EXPERIMENT

### A.3.1 RESULTS ON TRADITIONAL DATASETS

In addition to quantitative benchmarks, we further provide qualitative comparisons in Figure 4. We randomly selected two non-semantically manipulated images and five semantically manipulated images according to their proportions in the dataset, covering diverse object categories, background contexts, and manipulation types. This setup ensures a representative evaluation across both manip-

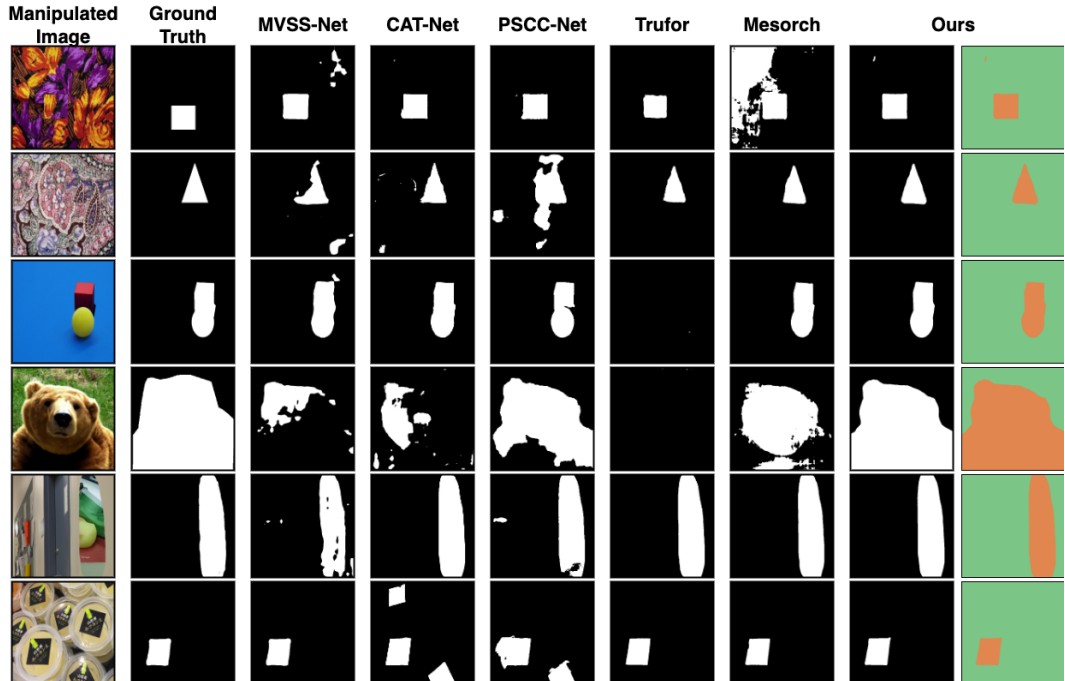

Figure 4: Qualitative analysis of SOTA models on conventional datasets. We randomly selected and compared two semantically manipulated images and five non-semantically manipulated images according to their proportions in the dataset. The first two rows show non-semantically manipulated examples, while the last four rows correspond to semantically manipulated cases. The rightmost column presents our two-step reasoning results: the orange region indicates Step 1, and the green region Step 2. The second-to-rightmost column shows the corresponding 0/1 mask outputs.

ulations with clear semantic meaning (e.g., replacing or altering salient objects) and manipulations that operate at a lower, often background or texture level without explicit semantic cues.

Compared with state-of-the-art baselines, our method produces more precise and coherent masks. Specifically, the predicted regions not only align with the manipulated object layout but also preserve fine-grained boundaries, even in challenging cases involving subtle splicing, occlusion, or background-level editing. For semantically manipulated examples, our model accurately captures the global structure of manipulated objects while suppressing false positives in unaltered areas, which is crucial for practical scenarios where semantic consistency is essential. In contrast, existing baselines often either over-segment (producing large false-positive regions) or under-segment (missing critical manipulated parts), leading to incomplete or noisy masks.

For non-semantically manipulated cases, where manipulations are less visually salient and often manifest as texture inconsistencies or geometric misalignments, our autoregressive paradigm demonstrates robustness by sequentially refining predictions and yielding compact, accurate masks that isolate the true tampered regions. Unlike conventional feed-forward architectures that may overlook weak or ambiguous traces, the autoregressive design enables iterative reasoning across spatial contexts, progressively consolidating local evidence into globally consistent predictions. Such a mechanism is particularly important in real-world images, where manipulations may be subtle and interwoven with natural variations.

Overall, these qualitative results confirm that our design generalizes well across manipulation types, successfully handling both semantically meaningful and background-level manipulations. They further illustrate that the autoregressive paradigm provides a principled way to enhance manipulation localization, leading to high-fidelity results and making our method better suited for deployment in diverse and unconstrained environments compared with prior approaches.

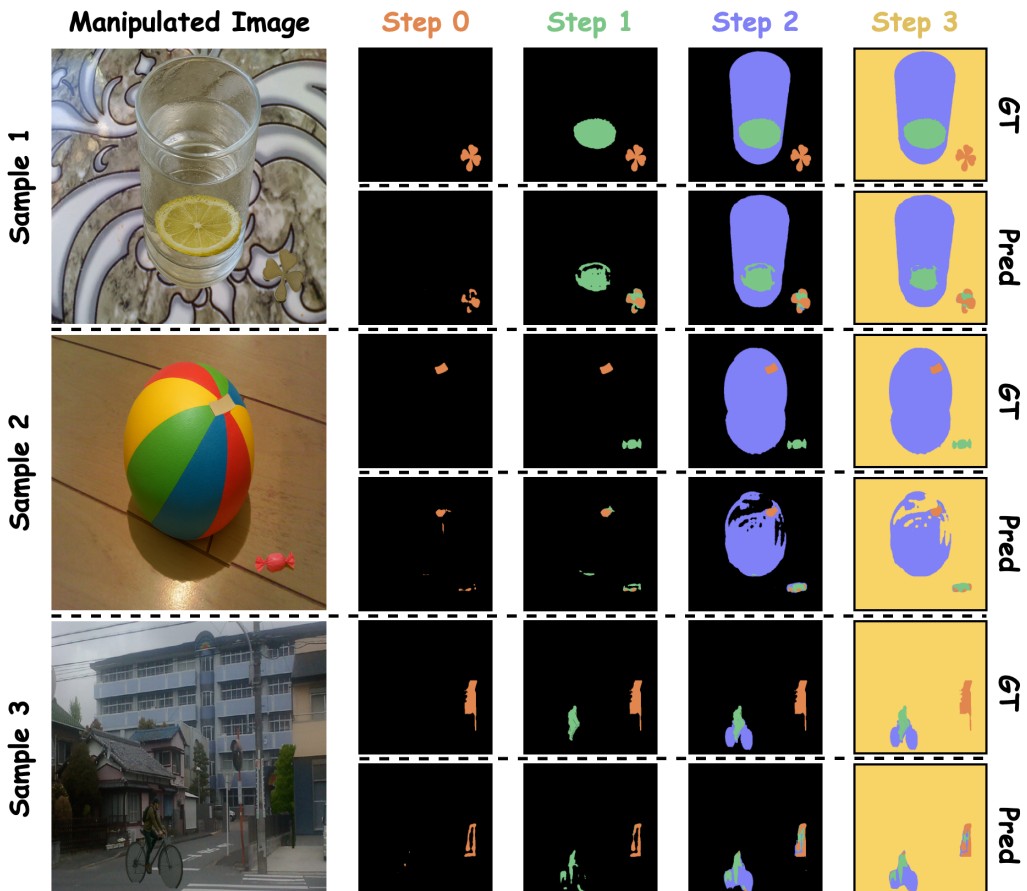

Figure 5: Qualitative results on the proposed HSIM dataset. Each row corresponds to one sample, where columns show sequential tampering steps (Step 0–Step 3). GT denotes the ground-truth mask at each stage, while Pred indicates our predictions. Our method successfully tracks manipulation evolution across steps, aligning predictions with the progressive nature of multi-step edits.

### A.3.2 RESULTS ON SEQUENCE MANIPULATION

To further evaluate the capability of our autoregressive framework, we conduct qualitative analysis on the proposed multi-step manipulation dataset, as shown in Figure 5. Unlike conventional benchmarks where manipulations are applied once, these cases involve sequential tampering operations, progressively altering different regions or objects in the same image. Such a setup better reflects real-world scenarios, where images may undergo multiple edits over time.

As illustrated in Figure 5, our method is able to trace manipulation evolution across steps, from the initial local insertion (Step 0) to cumulative object-level alterations (Step 3). The predicted masks closely align with ground-truth annotations at each stage, successfully distinguishing newly introduced manipulations from previously existing edits. This progressive localization ability highlights the effectiveness of our autoregressive paradigm: rather than collapsing all manipulations into a single mask, it incrementally builds up tampering evidence in a temporally consistent manner.

For example, in Sample 1, the small inserted pattern (Step 0) is correctly identified, followed by precise delineation of the lemon slice (Step 1) and cup boundary (Step 2). In Sample 2, where both object-level (ball) and small patch manipulations are present, our model captures the structural evolution while avoiding confusion between new and old tampering. Finally, in Sample 3, the method robustly localizes subtle manipulations across different semantic categories (e.g., structural edits on poles and bicycles), demonstrating generalization across diverse manipulation styles.

These results suggest that our framework is inherently well-suited for multi-step tampering detection, offering interpretability by revealing *how* manipulations accumulate and evolve, which is not possible with conventional one-shot segmentation baselines.

### A.4 LLM USAGE

In this paper, we use OpenAI's ChatGPT as a large language model (LLM) solely for general-purpose language refinement. Specifically, it was employed to polish grammar, improve readability, and ensure consistency of terminology across sections. No part of the research, including ideation, experimental design, implementation, or analysis, relied on ChatGPT. All conceptual contributions, technical designs, and empirical results are entirely the work of the authors.

