# OpenReview forum: "Does the Manipulation Process Matter? RITA: Reasoning Composite Image Manipulations via Reversely-Ordered Incremental-Transition Autoregression"
_ICLR.cc/2026/Conference — ICLR 2026 Conference Withdrawn Submission_

### Official Review · Reviewer_QeLE · 2025-10-26

**Soundness:** 2
**Presentation:** 1
**Contribution:** 1
**Rating:** 2
**Confidence:** 5

**Summary:**

The paper studies image manipulation localization. Rather than predicting all manipulation masks in a single-shot fashion, which suffers from the issue of dimensional collapse, the authors propose to model step-by-step manipulation processes in mask prediction. To do so, they reformulate image manipulation localization, construct a training set that demonstrates sequential and hierarchical manipulation of image contents, and design a neural model for the proposed task. The empirical results show that their model performs best under most setups.

**Strengths:**

- the paper reformulates image manipulation localization to emphasize sequentiality and hierarchy among manipulation steps
- a new model, training data, and evaluation metrics are introduced
- analysis and ablation studies are performed

**Weaknesses:**

- unclear motivation for localizing image manipulation; i expect to see it at the very beginning of the introduction.
- the related work section should comprehensively review existing works relevant to the task, model, data, and metrics, but currently, it simply repeats what has been said in the introduction section.
- the experimental results are mixed; the proposed model underperforms baselines under some evaluation setups, so it is unclear how effective it is.
- data curation relies on GPT-4o, but the potentially introduced biases are not discussed.
- most of the baselines are quite outdated, i.e., before 2023; thus, unclear how they would perform when equipped with the latest neural architectures and learning paradigms.
- claims are made without any references; for example, missing references to the claim in lines 45-46.
- lacking experiments on scaling up the size of training data; will the proposed paradigm remain better than baselines when both are trained on a large dataset, e.g., a dataset consisting of millions of examples?

**Questions:**

see above *Weaknesses*

- what are "undo" trajectories in line 195?
- can you please fix the incorrect citation styles, for example, in line 037?
- can you cite related work when using/mentioning existing resources, like CASIAv2 in line 170?

---

### Official Review · Reviewer_GCts · 2025-10-31

**Soundness:** 2
**Presentation:** 3
**Contribution:** 2
**Rating:** 4
**Confidence:** 4

**Summary:**

This paper introduces a novel paradigm for Image Manipulation Localization (IML), reformulating it from a conventional one-shot prediction task to a conditional sequence prediction task. The authors propose RITA, an autoregressive framework that predicts manipulation masks layer-by-layer, where each step is conditioned on the previous one. To support this new paradigm, the paper makes three core contributions: 1) the RITA framework itself, 2) a new real-world dataset, HSIM, with multi-step manipulation annotations, along with a method for synthesizing hierarchical data, and 3) a new evaluation metric, the HSS, designed to assess the accuracy of sequential and hierarchical predictions. Experiments demonstrate that RITA not only establishes a strong baseline on the new sequential task but also outperforms baselines.

**Strengths:**

-  RITA demonstrates state-of-the-art performance not only on its native sequential task but also on traditional one-shot benchmarks, where it outperforms existing methods in terms of both accuracy and efficiency

- The paper is written with great clarity.

**Weaknesses:**

- The paper's core evaluation relies on the HSIM dataset, which is generated exclusively by a single AI pipeline (GPT-Image-1). This raises significant concerns about source bias and generalizability. It is unclear if the model learns to detect fundamental manipulation traces or simply overfits to the specific artifacts of one generator. The evaluation should be more robust, testing the model against a diverse array of manipulation tools, including other state-of-the-art generative models (e.g., Qwen-Image, Nano Banana) that produce forgeries with far fewer artifacts.

- The definition of hierarchy is strictly limited to spatial containment. This overlooks more complex semantic hierarchies. Furthermore, both the synthetic and real-world HSIM datasets are generated using automated or AI-assisted processes, which may not capture the nuances, imperfections, and diverse techniques of manual, human-driven forgeries.

- It is difficult to determine whether the model's superior performance on traditional one-shot benchmarks stems from the novel sequential paradigm or simply from a more effective underlying architecture . This ambiguity makes it challenging to isolate and validate the paper's central claim that the sequential approach is inherently better for localization.

- Autoregressive models are inherently sequential and thus slower at inference time than parallel, one-shot models. The paper reports FLOPs for a single forward pass but does not discuss the total inference time, which would scale linearly with the number of manipulation steps. This could be a significant drawback in applications requiring rapid analysis.

**Questions:**

- Could you comment on the potential for source bias, as the HSIM dataset is created using only GPT-Image-1? How does RITA's performance generalize to forgeries from other modern generative models that may produce fewer artifacts?

- The paper defines hierarchy based on spatial containment. How would the framework handle more complex, non-nested manipulations that are semantically hierarchical?

- What is the total inference time compared to one-shot models, and what specific forensic scenarios justify this latency trade-off for knowing the edit sequence?

---

### Official Review · Reviewer_3bfE · 2025-11-01

**Soundness:** 3
**Presentation:** 2
**Contribution:** 3
**Rating:** 6
**Confidence:** 4

**Summary:**

The paper revisits image manipulation localization by framing it as a conditional sequence prediction problem, capturing how edits unfold step by step. The authors present RITA, an innovative model that explicitly captures the temporal and hierarchical nature of manipulation processes. The authors introduce HSIM, a real-world dataset simulating multi-step manipulations, and propose HSS, a tailored metric for evaluating hierarchical localization quality.

**Strengths:**

+ The paper introduces a new problem by reformulating manipulation localization as a conditional sequence prediction task, which allows the model to capture temporal and hierarchical relationships between editing steps.
+ Besides, the authors contribute a new dataset and evaluation metric tailored to this formulation.
+ The proposed RITA model shows good performance in experiments and ablations.

**Weaknesses:**

- When evaluating traditional one-shot benchmarks, the authors reformulate them as two-step tasks for RITA, while baselines operate in their native one-step mode. This raises fairness concerns and leaves it unclear whether the reported gains stem from the autoregressive design or from the task reformulation itself.
- Since RITA can theoretically run in a single step, adding a one-step version trained and tested under the original benchmark setup would clarify whether the improvements come from the model architecture itself.
- Moreover, the paper reports FLOPs and parameter counts for RITA but does not clarify whether efficiency is measured per prediction step or for full multi-step inference. This omission makes it difficult to fairly compare runtime with one-shot baselines, which perform only a single forward pass.
- In addition, although the paper provides one illustrative example of the manipulation process, the overall description of the HSIM dataset remains limited. Key details, such as data sources, operational distributions, and the proportion of manually refined samples, are missing, making it difficult to assess the dataset’s realism and reproducibility.

**Questions:**

1) In Figure 3, the “CGF Module” is shown in the model diagram, but its meaning is never explained in the text. Could you clarify whether CGF refers to the Transition Gated Fusion module? If so, please use a consistent name (either CGF or TGF) throughout the paper to avoid confusion.

2) In Figure 3.C, the Transition Gated Fusion module illustrates a cross-gating mechanism where the image feature is modulated by the mask feature and vice versa. However, Eq. (14) defines G_M = σ(W_I F_I), which seems inconsistent with the figure. Should this instead be G_M = σ(W_M F_M)? Please double-check the equations (also Eq. (13)) for consistency with the diagram and the intended cross-gating design.

3) Please clarify whether the reported FLOPs and efficiency metrics are measured per prediction step or across the full multi-step inference, and consider including total runtime and latency per image for a fair comparison with one-shot baselines.

4) The decoder fuses CGF outputs with mask features, but it is unclear how much each contributes to performance. An ablation comparing (1) CGF-only features and (2) CGF combined with mask or image features would clarify whether the gain stems from the fusion design or the additional feature inputs.

5) Since RITA can theoretically run with a single step, it would be helpful to include an experiment where a one-step version is trained and evaluated under the standard one-shot setting to clarify whether the performance gains arise from the autoregressive design or the model’s intrinsic architecture.

**Details Of Ethics Concerns:**

The HSIM dataset is described as being built upon CASIAv2 with additional manipulations generated using GPT-4o and GPT-Image-1, followed by manual refinements. Details about human involvement, compensation, and data release policies are missing. An ethics check is advised to ensure responsible data generation and release.

---

### Note · Authors · 2025-11-13

I have read and agree with the venue's withdrawal policy on behalf of myself and my co-authors.